# The New Kid on the Block: Online Adaptive Radiotherapy in the Treatment of Gynecologic Cancers

**Allen Yen, Chenyang Shen and Kevin Albuquerque \***

Department of Radiation Oncology, University of Texas Southwestern Medical Center, Dallas, TX 75390, USA
\* Correspondence: kevin.albuquerque@utsouthwestern.edu; Tel.: +001-214-645-8309

**Abstract:** Online adaptive radiation is a new and exciting modality of treatment for gynecologic cancers. Traditional radiation treatments deliver the same radiation plan to cancers with large margins. Improvements in imaging, technology, and artificial intelligence have made it possible to account for changes between treatments and improve the delivery of radiation. These advances can potentially lead to significant benefits in tumor coverage and normal tissue sparing. Gynecologic cancers can uniquely benefit from this technology due to the significant changes in bladder, bowel, and rectum between treatments as well as the changes in tumors commonly seen between treatments. Preliminary studies have shown that online adaptive radiation can maintain coverage of the tumor while sparing nearby organs. Given these potential benefits, numerous clinical trials are ongoing to investigate the clinical benefits of online adaptive radiotherapy. Despite the benefits, implementation of online adaptive radiotherapy requires significant clinical resources. Additionally, the timing and workflow for online adaptive radiotherapy is being optimized. In this review, we discuss the history and evolution of radiation techniques, the logistics and implementation of online adaptive radiation, and the potential benefits of online adaptive radiotherapy for gynecologic cancers.

**Keywords:** gynecologic cancer; online adaptive radiotherapy; radiation oncology

## 1. Introduction

Gynecologic (GYN) cancer is defined as any cancer originating in the women's reproductive organs. There are five major subtypes of GYN cancers including cervical cancer, uterine cancer, ovarian cancer, vaginal cancer, and vulvar cancer. About 94,000 women are diagnosed with gynecologic cancers in the US every year, with the most common GYN cancer being uterine cancer followed by ovarian cancer and cervical cancer [1].

The management of GYN cancers varies, but can involve surgery, chemotherapy, and/or radiation. Often, patients with locally advanced disease or high-risk features after surgery will receive either definitive radiation or adjuvant radiation while others may receive further radiation with brachytherapy. In this review, we will investigate the role of radiation in treating gynecologic cancers with a focus on the future role of online adaptive radiotherapy (ART) for GYN cancers, the next frontier in radiation therapy.

## 2. History of Radiotherapy for GYN Cancers

Radiation has long been used in the field of GYN cancers and the evolution of radiation in GYN cancers is shown in Figure 1. The first reports of radiation in GYN cancers came from the 19th century where physicians placed small pieces of radium near the cervix to treat cervical cancer [2]. In the 1960s these techniques were replaced with 2D radiation where patients were treated with external radiation based on bony landmarks using anterior-posterior beams or a four-field box with anterior-posterior/lateral beams. Using bony landmarks, the superior border was often placed at L4/L5 (or extended higher for para-aortic disease), the lateral borders were 1.5–2 cm lateral to the pelvis, and the inferior border was at least 3 cm below the lowest extent of vagina disease [3]. Unfortunately,

these traditional fields could often miss pelvic disease and treated a significant volume of uninvolved tissue.

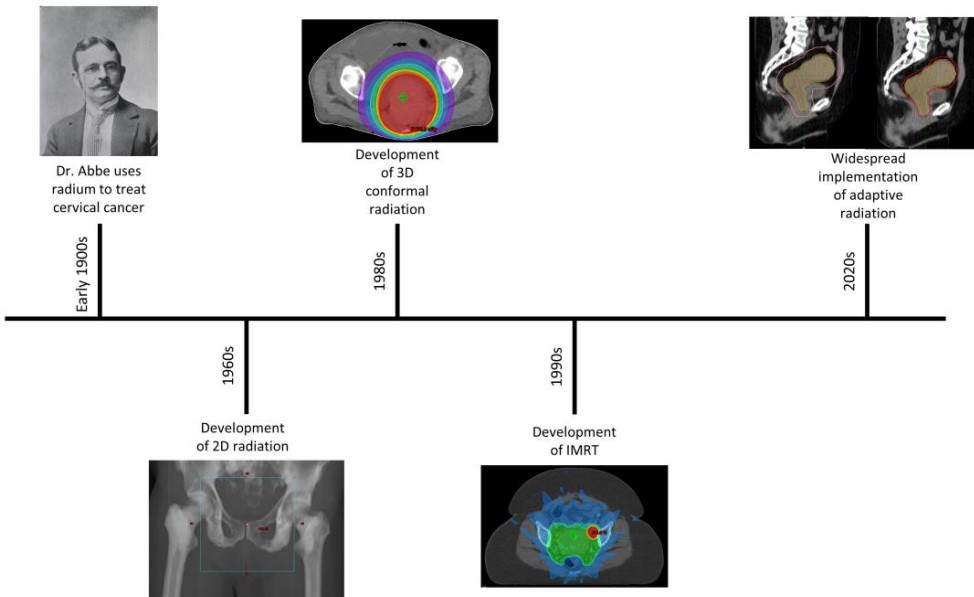

**Figure 1.** Timeline showing the evolution of radiation in the treatment of GYN cancers.

With the advent of 3D imaging using computed tomography (CT) and magnetic resonance imaging (MRI), radiation techniques have further improved with dose now delivered more precisely to targets while avoiding nearby organs-at-risk (OARs). Studies from the 1990s showed that treatment with 3D conformal radiotherapy improved target coverage while limiting dose to nearby organs such as the bladder and bowel [4]. With further refinement in radiation technology, intensity-modulated radiation therapy (IMRT) was developed. IMRT utilizes highly modulated radiation beams to better conform to 3D targets while sparing critical OARs. Multiple studies, including RTOG 1203, have shown that treatment with IMRT leads to significantly reduced GI and urinary toxicities for GYN cancers [5–7].

In the modern era of IMRT, patients with gynecological cancer who are planned for radiotherapy undergo a process called CT simulation. This step involves patients undergoing a CT scan, which radiation oncologists use to delineate targets and nearby normal structures in a process called contouring. Additional imaging with MRI and PET/CT can also help to better delineate structures. Targets including the cervix, uterus, vagina, and pelvic lymph nodes are contoured and constitute the gross tumor volumes (GTV) and clinical tumor volumes (CTV), which represent gross disease and microscopic disease, respectively. The CTV is further expanded to create a planning tumor volume (PTV). This expansion is conducted to account for errors during the delivery of radiation as well as potential movement of organs not only during treatment delivery, but throughout the course of fractionated treatments of radiotherapy. Nearby organs are also contoured including the rectum, bladder, sigmoid, bowels, femurs, and kidneys. An example of a GTV, CTV, and PTV is shown in Figure 2.

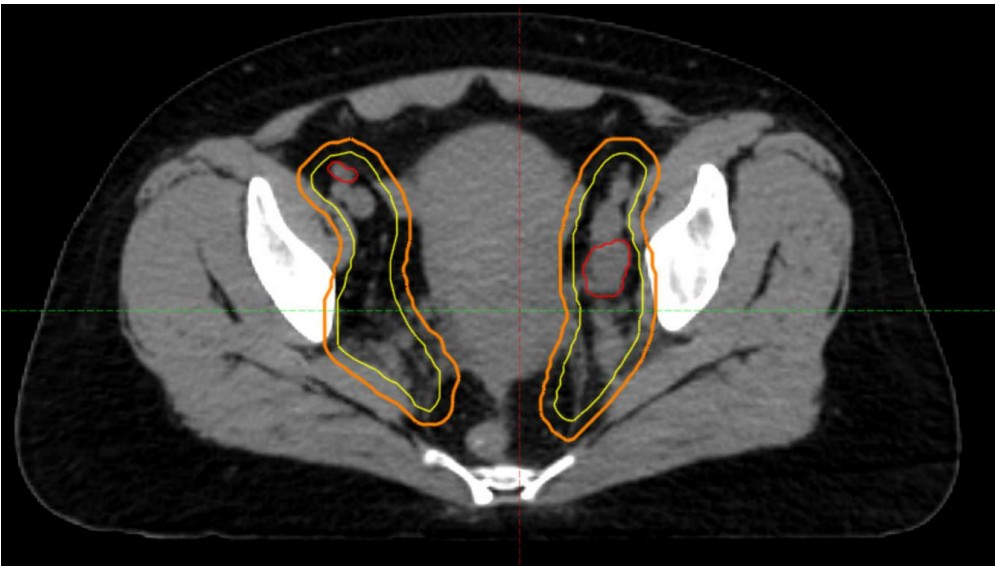

**Figure 2.** Representation of GTV, CTV, and PTV. A gross lymph node represents the GTV in red. The pelvic lymph node basin is the CTV in yellow with a 5 mm expansion to create the PTV in orange.

Treatment planning is particularly challenging for GYN cancers for several reasons. First, given the location of gynecologic cancers in the pelvis, changes in bladder filling and rectal gas can lead to significant motion. Previous studies in cervical cancer have shown that the interfractional motion (motion between treatments) of the cervix can vary up to 35 mm. In addition, patients undergoing definitive radiation treatment can have significant tumor regression that can further distort the expected anatomy of patients. Given these potential variations in anatomy, modern day clinical trials for cervical cancer recommend a CTV to PTV expansion of 1.5–2.0 cm [8].

With the targets and OARs defined, a radiation treatment plan is generated by trained dosimetrists. This plan is reviewed by the treating physician and if acceptable, is then delivered to the patient for all their planned fractions.

### 3. Adaptive Radiation Introduction

Adaptive radiation was first described by Yan et al. in the 1990s [9]. In this paper, the authors described a radiation treatment that could be modified during the course of treatment in response to changes in anatomy. This concept of adaptive radiation has become more popular recently with the recent development of linear accelerators (LINACs) dedicated for ART. Both cone-beam CT-based (CBCT) LINACs and MR-based LINACs have recently been developed and popularized for ART. Online ART is able to account for real-time changes in anatomy and adapt radiation plans during treatment delivery. Preliminary studies have shown that online ART can better spare OARs and improve target coverage in head and neck, abdomen, and pelvic cancers.

Prior to the development of these machines, there were less sophisticated methods for adaptive radiation including offline adaptive radiation treatments. With offline adaptation, patients underwent another CT simulation, re-contouring, and planning to account for changes in anatomy or tumor during treatment. Unfortunately, offline adaptive radiation is limited due to the high workload required to re-plan treatments. Another method to account for potential changes during radiation treatment is a plan of the day approach. This method was described by Bondar et al., where multiple treatment plans based on bladder volume are created and the plan utilized is determined based on the position of the CTV on cone-beam CT [10]. In their study, the PTV volume and volume of OARs inside the PTV were reduced.

Similarly, the advent of image-guided adaptive brachytherapy has made significant advances in the treatment of cervical cancer. The findings and recommendations from

EMBRACE I, EMBRACE II, and retroEMBRACE have revolutionized brachytherapy and have improved disease control and quality of life [11–13]. Online ART could lead to similar improvements.

GYN cancers can reap significant benefits from adaptive radiation for two major reasons. First, GYN cancers like cervical cancer can have significant changes in size over the course of treatments. Adaptive radiotherapy is able to account for these changes in tumor size and potentially treat less normal tissue while maintaining tumor coverage. Lastly, the pelvis contains the bladder, rectum, and bowel which can move significantly between treatment fractions. Again, adaptive radiation can account for this interfractional motion and potentially treat less normal tissue. These potential benefits are shown in Figure 3.

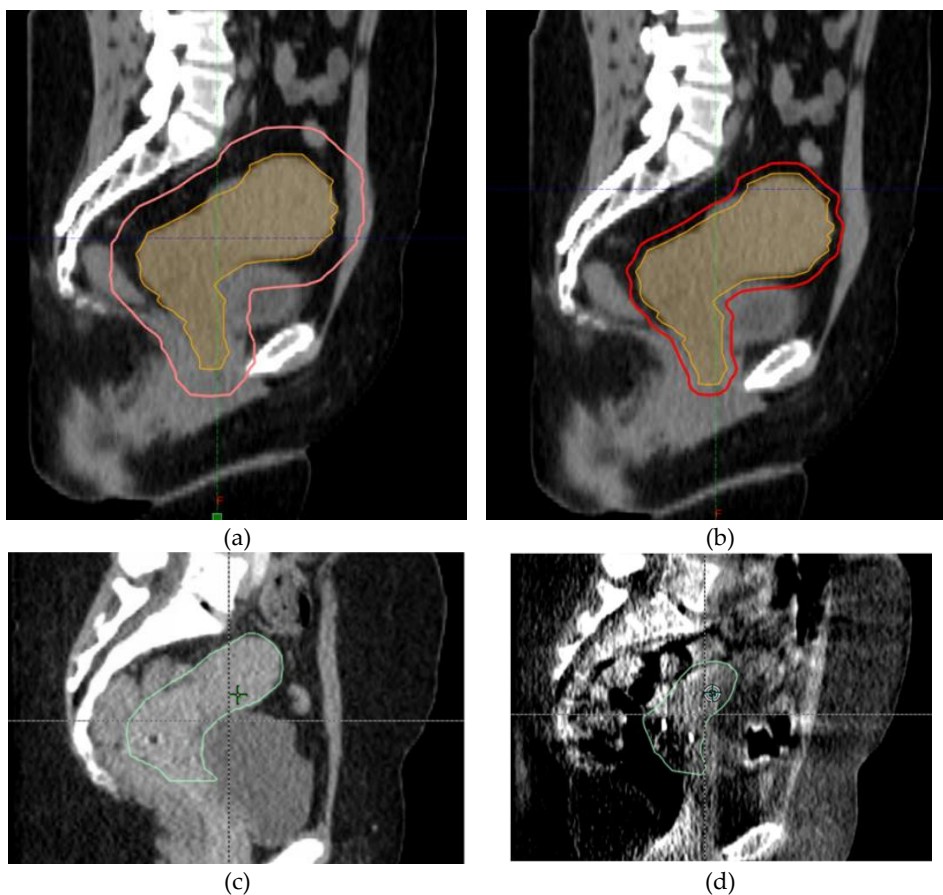

**Figure 3.** Potential benefits of ART. (**a**,**b**) show the decrease in CTV to PTV margins with daily adaptive radiation. The uterocervix CTV is in orange and the PTV is in red; (**c**,**d**) show the decrease in target size between the first fraction and last fraction. The uterocervix CTV is in green.

## 4. Logistics of Adaptive Radiation

### 4.1. CT Simulation

CT Simulation for patients undergoing adaptive radiotherapy for gynecologic cancers is the same compared to that of traditional non-adaptive treatments. Patients should be instructed to have a comfortably filled bladder and to perform an enema prior to CT Simulation. These are performed to create a reproducible bladder fill and to prevent the rectum from being expanded. Patients undergo imaging on a vac-lok bag (a moldable cushion used to reproduce patient's positioning consistently) in the supine position with the patient's arms on their chest or above their heads if para-aortic lymph nodes are involved. Some patients can also benefit from a marker in the vagina to help distinguish the location of the cervix or the uterine cuff. Intravascular (IV) contrast can further improve target and OAR delineation. Physicians can also consider additional imaging in the form of MRI or PET/CT.

### 4.2. Contouring

For the purposes of this review, we will not describe how to contour gynecologic cancers, but will refer readers to the variety of available resources for contouring [8,14–16].

### 4.3. Patient Selection

Patient selection is important for treating patients with gynecologic cancers with adaptive radiation. Given the novelty of this technology, the workflow for treating patients is still not fully optimized, and patients may be required to lie on the treatment table for a prolonged period. For this reason, patients with chronic back pain or those with para-aortic involvement who are required to have their arms above their head, may not be good candidates for adaptive radiation.

Furthermore, patients with complex disease/anatomy may also not be good candidates, i.e., para-aortic lymph node involvement or patients with significant surgical history. In our institution's experience, during the contouring process of online adaptive radiation, patients with complex anatomy may require additional time for contouring which can lead to more intrafractional motion (or motion during treatment). This intrafractional motion may require additional recontouring which can lead to a cycle of recontouring. For these patients, we would recommend either not treating with online adaptive radiation or utilizing larger CTV to PTV margins to account for potential intrafractional motion.

Lastly, patients with radioresistant disease (e.g., uterine sarcoma) may not be good candidates for online adaptive radiation. In these patients, their cancer may not change significantly, and they may not benefit from adaptive radiation.

While it is unknown which subset of patients would benefit the most from adaptive radiation, patients with disease that are expected to change significantly over time including advanced cervical cancers treated with definitive intent, gynecologic cancers with bulky nodal disease, inoperable endometrial cancers, large vulvar cancers, and tumors requiring re-irradiation will likely benefit, where adaptive radiation would allow tighter margins and greater normal tissue sparing.

### 4.4. Online Adaptive Radiation Workflow

The online ART workflow starts with on-board imaging aimed to obtain the anatomical/functional information on the day of treatment as guidance for adaptation. CBCT and MRI are the most common imaging modalities available in commercial on-line ART treatment units [17,18]. CBCT can be acquired quickly providing anatomical and physical properties of tissue in the scanned area. MRI, on the other hand, often takes longer to scan, but can offer better tissue delineation, and potentially some functional information [19]. Furthermore, functional imaging techniques, such as PET/CT, have also been integrated as an on-board imaging option which has the potential to provide additional information to guide online ART [20].

With images acquired, online contouring is performed to re-delineate treatment targets and critical organs to reflect the anatomical/functional changes of the patient at the time of treatment. Automatic contour generation is highly demanded to not only improve efficiency, but also to reduce manual efforts and potential human errors under intense time pressure. There are many approaches developed recently for automatic contour generation including artificial intelligence-based segmentation methods, which directly contour targets and OARs, and deformable registration-based methods, which propagates contours from the pre-planning stage to the images of the day. Regardless of the strategy employed for automatic contour generation, it is recommended that clinicians review and adjust contours if needed in routine clinical practice to ensure the integrity of contours.

With contours generated, online re-planning is then performed. Different commercial systems have implemented distinct plan optimization strategies to generate high-quality treatment plans efficiently.

Upon the completion of online re-planning, plan review and approval will be performed by the attending physician followed by treatment delivery. Unlike a conventional

workflow, patients remain on the couch throughout the entire ART session to minimize changes in patient positioning and anatomy. After plan review and approval, computation-based quality assurance (QA) is often performed using commercial or in-house software to ensure the proper delivery of treatment.

An example for a typical workflow for a CBCT-based system during the day of treatment is as follows: the radiation therapist brings the patient into the treatment vault and positions the patient accordingly. A cone-beam CT is acquired which is then deformably registered to initial planning CT and the treating physician is then paged to the machine. Next, the physician reviews the contours and OARs and adjusts as needed. PTV expansions and other derived structures are automatically generated based on the physician's contours. The dose is rapidly recalculated and a plan is automatically generated. The physician reviews this new plan and decides whether to treat with the new adapted plan or scheduled plan. Finally, QA is performed and the selected plan is delivered. Based on our institution's experience, the entire adaptive process takes ~40 min to treat a patient with pelvis-confined disease. The adaptive workflow is represented in Figure 4.

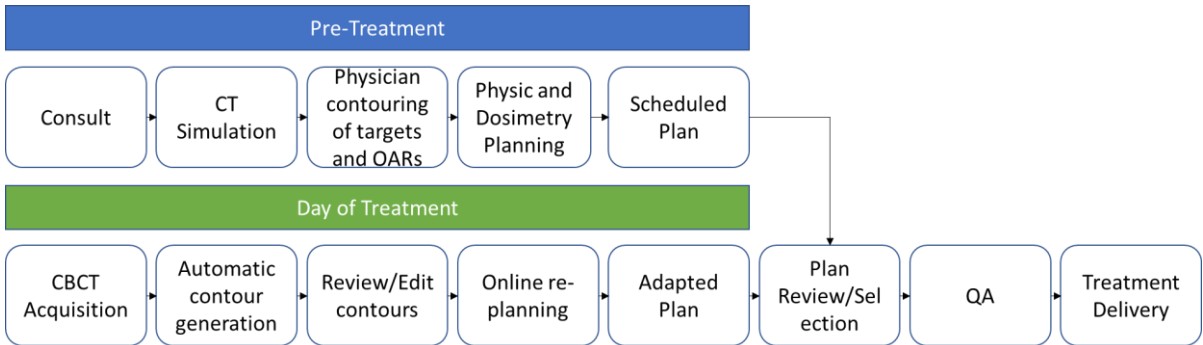

**Figure 4.** ART workflow. Prior to treatment, patients undergo consultation followed by CT Simulation. Physicians contour targets and OARs on the CT obtained, which is then given to a physicist/dosimetrist to create a radiation plan called the scheduled plan. On the day of treatment, patients undergo a CBCT. Contours are automatically generated and a physician reviews and edits contours as needed. A plan is generated in the TPS to create a plan called the adapted plan. The scheduled plan and adapted plan are reviewed by the physician and a plan is selected. QA is performed by the physicist and the selected treatment plan is delivered. OAR: organ-at-risk, CBCT: cone-beam CT, TPS: treatment planning system, QA: quality assurance.

### 4.5. Timing of Adaptation

With online adaptive radiation, adaptation can be scheduled in different ways including daily, weekly, or adapt-on-demand. As these names suggest, patients can undergo adaptations every day, weekly, or as needed.

The different timings of adaptive radiation can have a significant impact on clinic resources. Daily adaptive radiation requires the most resources and time while adapt-on-demand requires the least. On the other hand, more frequent adaptations can lead to improved dosimetry. This balance of improved dosimetry versus clinic resources as well as the optimal timing for adaptations is still under investigation. Potential factors that could impact the timing of adaptation include clinic resources available, location of the tumor, and the planned total dose.

### 4.6. Organ Motion/PTV Margins

Interfractional motion can be accounted for with online adaptive radiation and is one of the main benefits of online adaptive radiation for GYN cancers. However, physicians must still account for intrafractional motion. This is especially relevant for the bladder and rectum, which can lead to significant motion in the uterus and cervix. Previous reports have shown that changes in bladder filling can lead to intrafractional motion of the uterocervix

of up to 10.8 mm superiorly, 1.5 mm inferiorly, 3.19 mm anteriorly, 3.43 mm posteriorly, 2.74 mm to the left, and 2.48 mm to the right [21].

A recently published study described the potential for decreasing CTV to PTV margins from 1.5 to 2.0 cm down to 5 mm for patients undergoing daily adaptive radiation [22]. In this study, with a uniform 5 mm expansion, $98.39 \pm 2.97\%$ of the end-treatment CTV was covered in a validation cohort. Importantly, these patients were treated with only daily adaptive radiation. For patients undergoing weekly or adapt-on-demand, the study recommend larger 1.0–1.5 cm margins to account for potential interfractional motion during non-adapted fractions.

On the other hand, nodal volumes are often static and have minimal intrafractional motion. In a study by Bjoreland et al., they found that clinical nodal volumes had minimal motion in patients with prostate cancer [23].

## 5. Benefits and Limitations of ART for GYN Cancers

There are a number of potential benefits from online adaptive radiation. First, as described above, online adaptive radiation can potentially be used to reduce CTV to PTV margins. Based on the study described above, by reducing margins from 15 mm to 5 mm, the volume of bowel treated decreased by 292 cm$^3$. While this has not yet been proven to have a clinical benefit, there is an ongoing clinical trial investigating this [24]. This trial is investigating the benefits of ART to quality of life and toxicity improvements, while ensuring no compromise in tumor control and patient cancer outcomes.

Additionally, preliminary studies in other pelvic cancers have shown clear benefits in target coverage and OAR sparing with online ART. One such study, looking at on-line ART for prostate cancers showed a 13% increase in minimum prostate dose and a 13% decrease in dose to the rectum [25]. Another study looked at the benefits of an on-line adaptive plan-of-the-day approach for cervical cancers. In their study, if patients had >2.5 cm uterocervix motion, they had two plan-of-the-day plans generated. With these plans, they reported improved bowel doses by 26–29% [26].

Another potential benefit that has not been well studied is the utilization of adaptive radiation in response to biologic changes [11]. This may be the greatest benefit of ART. By monitoring tumor response to treatment at a physiologic or molecular level and tailoring treatment to those responses, we may be able to better utilize radiation in the treatment of GYN cancers. There are several on-going clinical trials examining the potential of biologic response driven ART [27–30]. Based on this notion, personalized ultrafractionated stereotactic adaptive radiotherapy (PULSAR) has become a new modality for delivering radiation where fractions, referred to as pulses, are delivered weeks apart with larger doses. In preliminary studies, immunotherapy in combination with PULSAR had better tumor control compared to traditional daily fractionation [31].

ART also comes with a number of limitations. The most obvious limitation is the significant required clinical resources required for the implementation of ART. This process involves physicians, physicists, dosimetrists, and therapists. Additionally, the time required for ART is also dependent on the accuracy of AI-segmentation and auto-planning. Lastly, online ART treatment machines can require significant economic investment for not only the purchase of the machine, but additionally maintenance.

## 6. Clinical Case

Our patient is a 36 year old African American women who initially presented with a 2 month history of vaginal bleeding. Pelvic examination showed a large cervical mass that was biopsied and confirmed to be an endocervical adenocarcinoma. Imaging with a pelvic MRI revealed an 8 cm cervical mass involving the lower uterine body and upper 2/3s of the vagina with bilateral and posterior parametrial invasion. PET/CT also showed the FDG-avid cervical mass with FDG-avid pelvic lymph nodes. She was staged as a FIGO stage IIIC1 cervical cancer and recommended for treatment with definitive concurrent chemoradiation.

Given the large size of her cervical mass, she was determined to be a good candidate for online ART as she was expected to have changes in the size of her tumor, and the reduced CTV to PTV margin could reduce the amount of normal tissue treated. She underwent daily online ART with a 5 mm CTV to PTV margin. She tolerated treatment well except for nausea that was controlled with oral medication. During the course of treatment, her uterocervix decreased in size significantly from 471 cm$^3$ to 191 cm$^3$ as shown in Figure 5. She completed her treatment with brachytherapy with tandem and ovoids. Unfortunately, she later presented with new pulmonary and liver lesions and is undergoing treatment with systemic therapy.

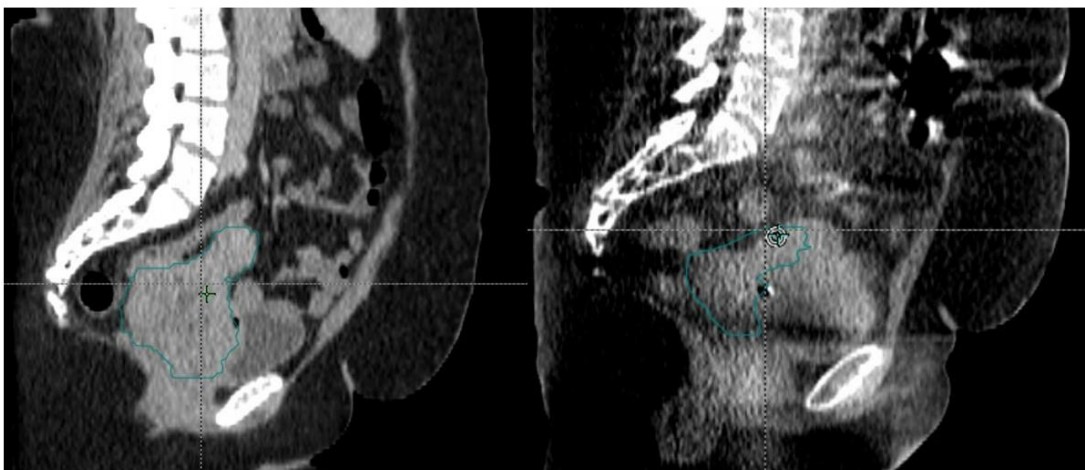

**Figure 5.** Change in size of uterocervix between fraction 1 and fraction 27 of our patient's radiation treatment course. The uterocervix CTV is in green.

## 7. Future Directions

Given the potential benefits of online adaptive radiation, numerous trials have been opened to explore the potential benefits. These trials are highlighted in Table 1.

**Table 1.** Ongoing clinical trials utilizing online ART.

| Clinical Trial Name | Goal |
|---|---|
| ARTIA-Cervix [24] | Demonstrate that ART for locally advanced cervical cancer will translate into decreased GI toxicities |
| Phase I Trial of Stereotactic MRI-Guided Online Adaptive Radiation Therapy (SMART) for the Treatment of Oligometastatic Ovarian Cancer [32] | Assess the feasibility of stereotactic MRI-guided online adaptive radiation therapy for treatment of oligometastatic ovarian cancer |
| Intratreatment FDG-PET During Radiation Therapy for Gynecologic and Gastrointestinal Cancers [33] | Evaluate the utility of adaptive intratreatment PET-CT planning for gynecologic and gastrointestinal cancers |

## 8. Conclusions

Online ART presents the newest frontier in the treatment of gynecologic cancers. The potential benefits from online ART include improved normal tissue sparing, improved target coverage, and improved treatment in response to biologic changes. However, they come at the cost of increased demand of clinic resources. These benefits need to be proven through clinical trials that are currently ongoing and recruiting. Online ART is the next step in the personalization of cancer care and has the possibility to revolutionize the treatment of GYN cancers.

**Author Contributions:** Conceptualization, A.Y. and K.A.; writing—original draft preparation, A.Y. and C.S.; writing—review and editing, A.Y., C.S. and K.A.; supervision, K.A. All authors have read and agreed to the published version of the manuscript.

**Funding:** This research received no external funding.

**Acknowledgments:** We would like to acknowledge the work of the many physicists and therapists who assisted us in the implementation of this new technology. We would also like to thank the patients who have been willing to undergo treatment with this new technology.

**Conflicts of Interest:** The authors declare no conflict of interest.

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
