# Peer review of "The New Kid on the Block: Online Adaptive Radiotherapy in the Treatment of Gynecologic Cancers"

_curroncol, doi:10.3390/curroncol30010066_

Round 1

Reviewer 1 Report

Dear Authors,

The topic is interesting, but I have some comments.

1. This title is in an american style,  what is interested for me and why I decided to review this article, but it could be different interpretate by others.

2. Abbreviation GYN is used correctly but not popular in other part of the world. I suggest "gynecological' not GYN.

3. What does it mean "online" in this situation? It is not described properly in my opinion.

4. I think that clinical case could be useful to better undestand the idea.

Reviewer 2 Report

This is a good general review of online adaptive radiotherapy in the treatment of gynecologic cancers.

The review in its current form gives a nice overview of the discussed issue. Most of the present literature regarding adaptive radiotherapy focuses on Cervix cancer. Additionally, adaptive brachytherapy gained all attention. Which might be relevant to refer to.

Patient selection should also give extra attention to patients who are curatively treated rather than postoperatively treated patients.

The images in figure 1 are better to be from gyn. patients.

The sentence in lines 196-197 needs to be more clarified.

4.4. Online Adaptive Radiation Workflow: this section should be more concise as it is well explained in figure 4.

Timing of adaption, the total / daily dose as well as the tumor location may play an important role in selecting the suitable protocol.

In lines 253-254 singular voice may be more appropriate, You referred to one ongoing study.

Lines 255 - 261, please rephrase!

Lines 262 - 267, please provide a reference!

Table 1, please give references if applicable!

You may add a paragraph regarding the limitations and the needed advances of AI in the field of RT such as AI-segmentation and auto-planning enabling adaptation within reasonable timeslots, training burden, economic consequences, and delay in machine timeslots.

Round 2

Reviewer 2 Report

Thank you very much